# Deep Molecular Response Achieved with Chemotherapy, Dasatinib and Interferon α in Patients with Lymphoid Blast Crisis of Chronic Myeloid Leukaemia

**DOI:** 10.3390/ijms24032050

**Published:** 2023-01-20

**Authors:** Lucia Vráblová, Vladimír Divoký, Pavla Kořalková, Kateřina Machová Poláková, Eva Kriegová, Romana Janská, Jan Grohmann, Milena Holzerová, Tomáš Papajík, Edgar Faber

**Affiliations:** 1Department of Haemato-Oncology, University Hospital Olomouc and Faculty of Medicine and Dentistry, Palacký University, 779 00 Olomouc, Czech Republic; 2Department of Biology, Faculty of Medicine and Dentistry, Palacký University, 779 00 Olomouc, Czech Republic; 3Institute of Haematology and Blood Transfusion, 120 00 Prague, Czech Republic; 4Department of Immunology, University Hospital Olomouc and Faculty of Medicine and Dentistry, Palacký University, 779 00 Olomouc, Czech Republic

**Keywords:** chronic myeloid leukaemia, blast crisis, tyrosine kinase inhibitor, interferon alpha

## Abstract

The treatment outcome in patients with chronic myeloid leukaemia (CML) in blast crisis (BC) is unsatisfactory despite the use of allogeneic stem cell transplantation (ASCT). Moreover, in some patients ASCT is contraindicated, with limited treatment options. We report the case series of two patients with lymphoid BC CML in whom ASCT was not approachable. The first patient developed BC two months after diagnosis in association with dic(7;9)(p11.2;p11.2) and T315I mutation. Blast crisis with central nervous system leukemic involvement and K611N mutation of the SETD2 gene developed abruptly in the second patient five years after ceasing treatment with nilotinib in major molecular response (MMR) at the patient’s request. Both underwent one course of chemotherapy in combination with rituximab and imatinib, followed by dasatinib and interferon α (INFα) treatment in the first and dasatinib alone in the second case. Deep molecular response (DMR; MR 4.0) was achieved within a short time in both cases. It is probable that DMR was caused by a specific immune response to CML cells, described in both agents. The challenging medical condition that prompted these case series, and the subsequent results, suggest a re-visit to the use of a combination of well-known drugs as an area for further investigation.

## 1. Introduction

As advances are made in the treatment of chronic phase chronic myeloid leukaemia (CML), fewer patients (approximately 0.7–4.5% percent at five years) are progressing to the accelerated phase or blast crisis (BC) [1,2,3]. The most recent WHO classification abandoned the term ‘accelerated phase’ while keeping in the definition of BC the same value for the blast count as in acute leukaemia (20%) [1]. In their latest guidelines, ELN experts still recommend the definition of BC with a threshold of 30% blasts; nevertheless, a recent ELN study on BC revealed a 20% threshold as more appropriate [2,4]. In approximately 30% of cases, the blast crisis in CML is of the lymphoid rather than myeloid phenotype [2]. The treatment outcome in patients with CML in BC is unsatisfactory [3,5]. According to recent studies, the combination of TKIs and chemotherapy improves the response rates and the survival of patients with lymphoid blast crisis [6,7,8,9]. Historical data from the pre-TKI era on a small cohort of patients with blast crisis on the achievement of haematological remission after interferon therapy are available [10]. The combination of dasatinib and IFNα suggests benefits compared with historical controls in small single-arm studies as a first-line therapy; however, this treatment option was never tried in patients with BC [11,12]. In general, an attempt should be made to return the patient to a chronic phase in order to proceed with allogeneic stem cell transplantation (ASCT). There is a higher probability of successful ASCT in the second chronic phase than in overt BC; however, the risk of failure or relapse after ASCT is considerably higher [13,14]. The prognosis for patients who cannot receive a transplant is dismal [3,4,5]. We describe here the case series of two patients with lymphoid BC CML in whom ASCT was not approachable.

## 2. Results

### 2.1. Case Report 1

A 51-year-old man was diagnosed with chronic phase CML in April 2014. The Sokal, EUTOS, and ELTS scores indicated a low-risk disease. Chromosomal analysis revealed a Philadelphia chromosome and additional dic(7;9)(p11.2;p11.2) chromosomal abnormality. The retrospective analysis of mutations in the ABL kinase domain by next-generation sequencing (NGS) did not detect any pathogenic mutation at the time of diagnosis.

Nilotinib was initiated with a dose of 600 mg daily in April 2014 and complete haematological response (CHR) was achieved within the first month of treatment. However, two months later the patient developed lymphoid BC, with 40% B-lymphoblasts in peripheral blood confirmed by flow cytometry. Chromosomal analysis did not reveal any clonal evolution, however, but this time NGS detected p.T315I mutation. The patient underwent induction therapy according to the modified GMALL protocol consisting of daunoblastine (80 mg per day; days 7–8 and 14–15 of chemotherapy), vincristine (2 mg per day; days 7 and 14 of chemotherapy), cyclophosphamide (360 mg per day; days 4–6 of chemotherapy), and asparaginase (9000IE; days 17 and 25 of chemotherapy) with additional rituximab (100 mg day 1, 200 mg day 2, 400 mg day 3 of chemotherapy), followed by imatinib 400 mg, as no other TKI could be used with respect to the health insurance company limitation. Liquor analysis did not confirm central nervous system (CNS) leukemic involvement and the prevention of CNS disease was performed with the intrathecal application of standard cytarabine, methotrexate, and dexamethasone doses on the 2nd, 31st, and 33rd days of chemotherapy. Complete haematological response and cytogenetic response (CCyR) were achieved on day 28 after initiation of the chemotherapy, but without significant molecular response (RT-Q-PCR 0.18% IS, Figure 1). During and shortly after chemotherapy, several serious complications occurred: hepatopathy (grade 3), severe mucositis (grade 3), ileus (grade 3) treated conservatively without surgical intervention, cerebral toxoplasmosis diagnosed by an MRI scan and the positive response to the anti-toxoplasmosis treatment, drug-induced acute pancreatitis (grade 3), and possible fungal pneumonia detected on a high-resolution CT scan. During this period, the patient lost 12 kg of body weight and his Karnofsky performance score decreased to 50. The RT-Q-PCR for BCR::ABL1 progressed to 9.26% IS at day 84 after the start of induction chemotherapy, but p.T135I was not detected any more. Dasatinib was initiated with a dose of 40 mg per day five times per week. Because of severe neutropenia (grade 3), the patient had to discontinue dasatinib for two weeks and also received G-CSF 48 ug 13 times over eight weeks. After the correction of neutropenia, 40 mg dasatinib was reintroduced twice weekly and after the resolution of neutropenia three times weekly. In March 2015, bone marrow aspiration confirmed continuous CCyR and, surprisingly, the achievement of a major molecular response (MMR) with the BCR::ABL1/ABL1 level 0.08% IS. To prevent the development of resistance to dasatinib, and because of the impossibility of proceeding with ASCT because of serious complications of the chemotherapy, interferon alpha (INFα) was added at an initial dose of 3MIU twice weekly in March 2015. Fifteen months after the lymphoid BC, MR 5.0 was achieved, while the dose of dasatinib was 40 mg per day and INFα was given 3MIU three times weekly. Three months later, the patient was newly diagnosed with nephrotic syndrome and moderate renal insufficiency as a side effect of IFNα, which was stopped immediately. The daily 40 mg dose of dasatinib was maintained. The patient died of influenza pneumonia in January 2020, maintaining MR 5.0 (67 months after the initiation of the treatment for BC).

### 2.2. Case Report 2

A 54-year-old man was diagnosed with chronic phase CML in October 2010. The Sokal, EUTOS, and ELTS scores indicated a low-risk disease. Chromosomal analysis was not performed initially because of the patient’s refusal. The treatment was initiated with imatinib 400 mg daily and changed two months later to nilotinib 600 mg daily after enrolment into the clinical trial CAMN107EIC01 with nilotinib. Cytogenetic response was achieved within four months of treatment, and MMR within six months. It should be mentioned that the entire treatment of the patient was significantly influenced by his poor compliance. In June 2015, the patient decided to interrupt nilotinib, when his BCR-ABL1 level was 0.05% IS. The patient had never achieved DMR before the interruption of treatment. After the drug had been stopped, BCR::ABL1 monitoring was performed every month for one year, and after that one more check was performed one year later. During this period, no molecular recurrence was detected; the last BCR::ABL1 measured in June 2017 was 0.02% IS; afterwards, he ceased to respect the regular visit plan. In April 2020, the patient was examined at a local hospital because of thrombocytopenia, and in our centre we verified lymphoid BC with 75% B-lymphoblasts in the bone marrow as assessed by flow cytometry. Chromosomal analysis did not reveal any further clonal evolution; NGS detected K611N mutation of the SETD2 gene. The patient underwent reduced therapy in one day consisting of rituximab (750 mg), cyclophosphamide (1200 mg), vincristine (2 mg), and prednisone (80 mg per day; days 1–5 of chemotherapy), followed by imatinib 400 mg daily. As liquor analysis confirmed CNS leukemic involvement, the intrathecal application of standard doses of cytarabine, methotrexate, and dexamethasone on the 3rd, 8th, and 11th days of chemotherapy was performed. Dasatinib at 100 mg daily dosage was initiated on day 10 after the chemotherapy. Deep molecular response was surprisingly achieved on day 20 after the initiation of the chemotherapy (RT-Q-PCR 0.002% IS, Figure 2). In order to prevent the development of resistance to dasatinib, the possibility of ASCT, intensive chemotherapy, or at least additional INFα, was repeatedly discussed; however, the patient refused any other option aside from dasatinib, arguing that the addition of chemotherapy would interfere with his immune system and cause unbearable risk during the COVID-19 epidemic. The patient lost MR 5.0 in November 2020 (eight months after BC) with the confirmation of p.T315I mutation by NGS. Ponatinib 45 mg daily was started, with the intermediate response (MR 4.0) achieved two months later. In February 2021, the patient lost MMR (RT-Q-PCR 7.87% IS). Bone marrow aspiration was provided and verified a second blast crisis with 60% lymphoblasts and cytogenetic evolution. The patient was offered various therapeutic options (chemotherapy, the addition of interferon to TKIs); unfortunately, he refused all of them. The patient died five months later because of disease progression complicated by severe infection (fifteen months after the diagnosis of BC).

## 3. Discussion

The probability of BC during the initial phase of nilotinib treatment has been reported to be 1% in the ENEST trial [15]. Both patients achieved an optimal treatment response to the initial therapy; however, progression to the BC occurred in a different way. The first patient progressed very early as a result of mutation in the kinase domain of BCR::ABL1 and additional cytogenetic abnormality dic(7;9)(p11.2;p11.2). The BC of the other patient occurred after a lengthy treatment-free remission after the achievement of MMR, and no abnormality other than K611N mutation of the SETD2 gene was detected. Current recommendations refer to abnormalities of chromosome 7 as -7/7q- as high-risk additional chromosomal aberrations (ACAs) predicting a poorer response to TKIs and a higher risk of progression [16,17,18,19]. The significance of a structural change of chromosome 7 presented as a dicentric chromosome is unknown [2]. Also, in our patient, the most common mutation p.T315I in the ABL kinase domain was detected during the development of BC, which has been associated with TKI resistance to first- and second-generation TKIs [2,5]. Relapse events occurring within 12 months after TKI discontinuation are rare; however, the EURO-SKI and STIM study groups reported late molecular relapses beyond 24 months [20,21]. The inability to perform regular monitoring associated with patient non-compliance resulted in the patient being diagnosed with BC without previous warning five years after the interruption of TKI therapy. There are known positive factors for TFR after TKI discontinuation, such as the duration of TKI, the duration of the response, and the depth of molecular response [20]. Besides the patient’s poor compliance, an obvious reason for his progression could be that he had never achieved DMR before treatment interruption [20,21]; however, the detection of the SETD2 gene mutation could be a significant immunological trigger for the late relapse of the disease because of the loss of the tumour suppressor gene function that causes genome instability, apoptosis inhibition, and the loss of cell cycle control [22].

Currently, more treatment options are available than we were able to provide to the first patient in the past. However, the treatment of rare and heterogenous BC-CML remains challenging also for the difficulty of designing clinical trials in this setting [3,5]. Unfortunately, for various reasons, ASCT could not be performed on our patients. The reason why the ponatinib treatment was not initiated with the presence of the T315I mutation in the first patient was that, at the time of the patient’s progression, dasatinib was the only other available treatment after the failure of nilotinib. Being worried about the high risk of the development of resistance to the administration of dasatinib, we added IFNα to the treatment with the aim of targeting leukemic stem cells (LSCs) and also covering the BCR::ABL1 independent ACA dependent mechanisms of possible progression. Tyrosine kinase inhibitors do not appear to eliminate CML stem cells, and attempts to discontinue therapy have usually resulted in relapse, while IFNα may well be able to target LSCs responsible for disease initiation via cytostatic and immunomodulatory effects. The mechanism of action of IFN-α therapy is incompletely understood. The IFN-α treatment may induce cycling and thereby expose LSCs to the effects of TKIs and chemotherapeutic agents, and also induce the recognition and elimination of CML cells by the immune system. Patients who have been successfully treated with IFN-α express increased amounts of cytotoxic T lymphocytes (CD8+ T-cells) and natural killer (NK) cells. The immunostimulatory effect of dasatinib was also described, caused by its ability to induce the expansion of large granular lymphocytes (LGLs), consisting of mono- or oligo-clonal CD8+ T cells and NK cells, shown to correlate with better prognosis and consequently a more favourable response in patients with CML. The combination of these two drugs may therefore be advantageous [23,24,25,26,27]. Because of the low dosing of dasatinib in our patient, we did not observe any consistent long-term changes in the counts of lymphocytes and/or their subpopulations, such as T lymphocytes or NK cells. Unfortunately, we have not been able to perform a more extensive examination of other immune changes, such as those described in patients with long-term molecular remission after interferon [28]. The German CML and Scandinavian CML research groups have shown the potential of a deeper response in high-risk CML by adding IFN-α to imatinib [29]. These claims are confirmed by our first patient, in whom the potentiation of the therapy by INF-α was able to sustain a prolonged response.

## 4. Conclusions

Our case series highlights the long-term deep molecular response achieved with the combination of dasatinib with INFα after the abrupt lymphoblastic BC of CML. We believe that the treatment effect was achieved with the immune mechanisms of both drugs, thus deserving further investigation, taking the mutations found into account. In addition to this, we report the very uncommon case of abrupt BC evolving late after stopping the tyrosine kinase inhibitor in the molecular response.

## Figures and Tables

**Figure 1 ijms-24-02050-f001:**
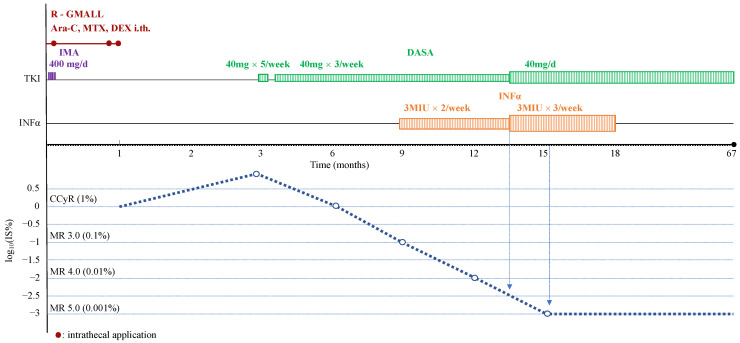
Clinical course of the blast crisis of Patient 1. Abbreviation: R-GMALL, rituximab-German multicentre acute lymphoblastic leukaemia protocol; Ara-C, cytosine arabinosine; MTX, methotrexate; DEX, dexamethasone; i.th., intrathecal; IMA, imatinib; DASA, dasatinib; TKI, tyrosine kinase inhibitor; INFα, interferon α; CCyR, complete cytogenetic response; MR, molecular response.

**Figure 2 ijms-24-02050-f002:**
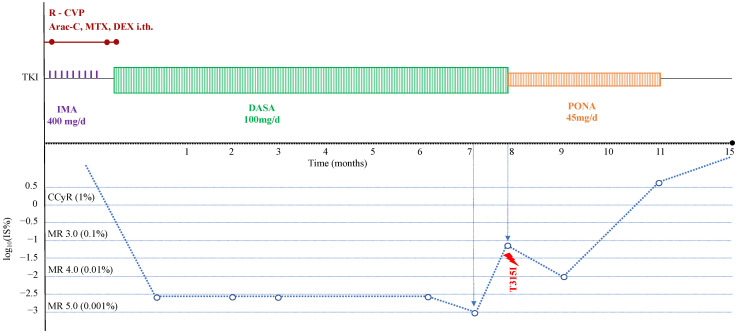
Clinical course of the blast crisis of Patient 2. Abbreviation: R-CVP, rituximab-cyclophosphamide, vincristine, prednisone; Ara-C, cytosine arabinosine; MTX, methotrexate; DEX, dexamethasone; i.th., intrathecal; IMA, imatinib; DASA, dasatinib; PONA, ponatinib; TKI, tyrosine kinase inhibitor; CCyR, complete cytogenetic response; MR, molecular response.

## Data Availability

The data presented in this study are available on request from the corresponding author. The data are not publicly available due to privacy issues.

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
