# Peer review of "Deep Molecular Response Achieved with Chemotherapy, Dasatinib and Interferon α in Patients with Lymphoid Blast Crisis of Chronic Myeloid Leukaemia"

_ijms, 2023, doi:10.3390/ijms24032050_

Round 1

Reviewer 1 Report

The cases presented by the authors are clear and very well detailed. They showed a high management of the leukemic disease and treatments.

Minor recommendations:

1. The authors should include some statements about the general treatment and the use of Interferon α during BC, since this point was included in the title and I understand that it is the central point of the discussion.

2. In the discussion and conclusion they mentioned that ..."the effect of the treatment was achieved with the immune mechanisms of both drugs". It will be good to briefly mention what are the actions of the immune mechanism.

3. The sentence on line 66 "400mg administered due to insurance restrictions" appears to be meaningless and should be rewritten.

Author Response

Manuscript ijms-2156535

Response to Reviewers

Reviewers' Comments to the Authors:

The cases presented by the authors are clear and very well detailed. They showed a high management of the leukemic disease and treatments.

Thank you. We appreciate you for your precious time in reviewing our paper and providing valuable comments.

  1. The authors should include some statements about the general treatment and the use of Interferon α during BC, since this point was included in the title and I understand that it is the central point of the discussion.

Author response: Thank you for pointing this out. This is correct, and we have expanded the introduction of manuscript including general statement of treatment of blast crisis of chronic myeloid leukaemia focusing on INFα.

  1. In the discussion and conclusion they mentioned that ..."the effect of the treatment was achieved with the immune mechanisms of both drugs". It will be good to briefly mention what are the actions of the immune mechanism.

Author response: As suggested, a brief statement of mechanism of actions of dasatinib and INFα in CML immune system was added to the discussion.

  1. The sentence on line 66 "400mg administered due to insurance restrictions" appears to be meaningless and should be rewritten.

Author response: The sentence on line 66 was rewritten to be more meaningful.

Reviewer 2 Report

This is a small case cohort of just two patients who have developed blast crisis post TKI treatment.   The responses using IFN are interesting and relevant given the low incidence of the development of blast crisis.

The report is slightly long considering the number of patients.  

Author Response

Manuscript ijms-2156535

Response to Reviewers

Reviewers' Comments to the Authors:

This is a small case cohort of just two patients who have developed blast crisis post TKI treatment.   The responses using IFN are interesting and relevant given the low incidence of the development of blast crisis.

Thank you. We appreciate you for your precious time in reviewing our paper.

The report is slightly long considering the number of patients.

Author response: The cases presented tried to be well-detailed with no space for shorting, however the discussion was slightly shortened as the information are now presented in the introduction respecting comments of second reviewer.